# The Relationship between Vitamin K and Osteoarthritis: A Review of Current Evidence

**DOI:** 10.3390/nu12051208

**Published:** 2020-04-25

**Authors:** Kok-Yong Chin

**Affiliations:** Department of Pharmacology, Faculty of Medicine, Universiti Kebangsaan Malaysia, Cheras 56000, Malaysia; chinkokyong@ppukm.ukm.edu.my; Tel.: +603-9145-9573

**Keywords:** carboxylation, cartilage, joint, menaquinone, phylloquinone

## Abstract

Vitamin K is a cofactor of γ-glutamyl carboxylase, which plays an important role in the activation of γ-carboxyglutamate (gla)-containing proteins that negatively regulate calcification. Thus, vitamin K status might be associated with osteoarthritis (OA), in which cartilage calcification plays a role in the pathogenesis of the disease. This review collates the evidence on the relationship between vitamin K status (circulating or dietary intake level of vitamin K, or circulating uncarboxylated gla proteins) and OA from human observational studies and clinical trial, to examine its potential as an agent in preventing OA. The current literature generally agrees that a sufficient level of vitamin K is associated with a lower risk of OA and pathological joint features. However, evidence from clinical trials is limited. Mechanistic study shows that vitamin K activates matrix gla proteins that inhibit bone morphogenetic protein-mediated cartilage calcification. Gla-rich proteins also inhibit inflammatory cascade in monocytic cell lines, but this function might be independent of vitamin K-carboxylation. Although the current data are insufficient to establish the optimal dose of vitamin K to prevent OA, ensuring sufficient dietary intake seems to protect the elderly from OA.

## 1. Introduction

Osteoarthritis (OA) is a debilitating disease of the movable joints commonly experienced by the elderly. It is characterised by deterioration of the articular cartilage, alteration of subchondral bone, formation of osteophytes, joint space narrowing, and inflammation of the synovium [1]. The Global Burden of Disease 2010 study reported that the age-standardized prevalence of hip OA was 0.85% (95% uncertainty interval (UI) 0.74% to 1.02%) and for knee OA, it was 3.8% (95% uncertainty interval (UI) 3.6% to 4.1%). This was accompanied by a year-lived-with-disability count of 17.1 million in 2010 [2]. With the increasing elderly and obese populations worldwide, the prevalence of OA is expected to increase because age and obesity are major contributing factors of OA, besides being female and having experienced joint trauma [3,4]. 

The current OA pharmacotherapies, such as paracetamol, non-steroidal anti-inflammatory agents (NSAIDs), and tramadol, focus on pain amelioration but do not solve the underlying pathologies of OA [5]. Numerous preventive agents for OA are available in the market, such as glucosamine sulfate, chondroitin sulfate, curcumin, and avocado soybean unsaponifiables but their efficacy remains elusive [6,7,8]. Dietary components, such as vitamin D, vitamin E, polyphenols, have been reported to retard the progression of OA [9,10,11,12].

Cartilage calcification plays an important role in the pathogenesis of OA. Calcification of the articular cartilage at the hip and knee is highly prevalent among the general population [13]. The presence of calcification was reported to correlate with the histological score of OA at the hip and knee [13]. In animal studies, cartilage calcification has been associated with decreased compressive strength and increased friction of the cartilage [14,15]. Additionally, the presence of basic calcium phosphate crystals has been shown to induce proliferation of synovial fibroblasts and increase the expression of metalloproteinase-13 in chondrocytes [16]. Thus, cartilage calcification in tandem with increased expression of tumor necrosis factor-alpha and interleukins from macrophages induced by wear-and-tear particles could lead to further cartilage breakdown and OA. 

Matrix gamma-carboxyglutamic acid (gla) protein (MGP) involved in calcification is expressed in chondrocytes, the cartilage forming cells. MGP knockout animals developed spontaneous calcification of the blood vessels [17], while overexpression of MGP retarded the mineralization of bone [18]. MGP acts as an inhibitor to bone morphogenetic protein-2 mediated calcification [19] and polymorphism of MGP has been associated with OA [20]. The biological activity of MGP is dependent on vitamin K, a cofactor to the enzyme γ-glutamyl carboxylase that converts the inactive uncarboxylated MGP to the active carboxylated MGP. Thus, the carboxylation status of MGP can be used to assess the vitamin K status of an individual [21]. 

Vitamin K can be divided into phylloquinone (vitamin K1) with a phytyl group obtained from plants and menaquinones (vitamin K2) with a polyisoprenyl side chain with 6 to 13 isoprenyl units at the 3-position, formed by lower bowel bacteria. Apart from the aforementioned MGP, vitamin K also facilitates the carboxylation of other proteins, such as gla-rich protein (GRP) and osteocalcin (both of which are vital in regulating calcification), and prothrombin, which is important in regulating the coagulation cascade [22,23]. 

It is tempting to speculate that vitamin K, a dietary component, can influence cartilage calcification through carboxylation of MGP, and prevent the occurrence of OA. Therefore, a literature search was performed to identify evidence on the relationship between vitamin K status and OA. The evidence was summarized in this review to provide a comprehensive view of the potential of vitamin K as an agent to prevent the progression of OA. Human studies were presented and discussed according to the strength of evidence, followed by a discourse on the mechanism of action of vitamin K in preventing OA.

## 2. The Relationship Ship between Vitamin K Status and OA

### 2.1. Case-Control Studies

The relationship between vitamin K and OA have been explored in several case-control studies sampling cartilage from OA patients prior to surgery or organ donors. An earlier study by Roberts et al. [24] reported that serum phylloquinone level, but not carboxylated/undercarboxylated osteocalcin, was significantly lower in elderly patients with femoral neck fracture and OA (n = 13; median age ≥ 80 years) compared to healthy control (n = 16; median age ≥ 71 years), suggesting that it is a potential marker of skeletal health. However, serum phylloquinone level was not significantly different between patients with femoral neck fracture and OA per se. One day after hip replacement surgery, the serum phylloquinone level decreased to non-detectable levels and rebounded after one week, suggesting that vitamin K might be involved in the skeletal healing process [24]. In a more recent study, plasma phylloquinone level was found to be lower in patients with knee OA (n = 40, aged 50.4 ± 4.9 years) compared to healthy control (aged 48.9 ± 4.6 years). Increased plasma phylloquinone level was linked to increased medial cartilage thickness. Subjects with vitamin K deficiency was found to have higher scores of Western Ontario McMaster Scale (WOMAC), which reflects an increased severity of the disease [25].

Other studies used carboxylation of MGP or gla-rich protein (GRP) as a marker of vitamin K status in patients, but its relationship with OA was heterogeneous. Serum uncarboxylated MGP was reported to be higher in the non-arthritic controls (n = 30; median age 42 years), compared to patients with arthritis. Synovial uncarboxylated MGP was higher in patients with inflammatory arthritis (n = 9; median age 55 years; juvenile idiopathic arthritis and ankylosing spondylitis) compared to patients with non-inflammatory arthritis (n = 17; median age 55 years; OA and chondracalcinosis), but the inverse was found for serum uncarboxylated MGP. Overall, synovial/serum uncarboxylated MGP ratio was higher in the inflammatory arthritis group compared to the non-inflammatory group [26]. Immunohistology staining of cartilage samples harvested from OA patients undergoing total knee replacement surgery showed that more undercarboxylated GRP was found in the cartilage matrix and synovial membrane of OA patients, compared to the carboxylated form. The findings suggest that uncarboxylated MGP leaking to the systemic circulation could be an index of discrimination between OA and non-OA conditions.

In contrast, serum uncarboxylated MGP was reported to be lower in patients with knee OA (n = 178; mean age 62.8 ± 7.4 years) than normal controls (n = 160; mean age 63.2 ± 8 years), according to Bing et al. [27]. The synovial uncarboxylated MGP also correlated negatively with disease severity among OA patients [27]. The reason for this deviation with previous studies was not explained by the authors.

The findings of case-control studies have been summarized in Table 1.

### 2.2. Cross-Sectional Studies

The relationship between vitamin K status (direct measurement or matrix protein carboxylation) and occurrence or progression of OA has been addressed in several studies. In the Framingham Offspring Study (n = 672; mean age 66 years), a negative relationship between the occurrence of hand OA, large osteophyte and joint space narrowing and circulating phylloquinone was found. The level of 1 nmol/L circulating phylloquinone level was identified as the threshold level, whereby further increment of circulating phylloquinone would not enhance the prevention of OA. In addition, knee OA was not significantly associated with circulating phylloquinone in this study [29]. However, this study did not include carboxylation as part of the variables. Ishii et al. [30] reported that menaquinone level was significantly higher in the femoral/tibial lateral condyles, compared to medial condyles among patients (n = 58; mean age 73 ± 8 years) with Kellgren-Lawrence (KL) grade 4 undergoing total knee arthroplasty, regardless of age and sex. This observation was possibly contributed by the more severe OA condition at the medial condyles, compared to the lateral side. 

With regards to dietary intake, Research on Osteoarthritis Against Disease (ROAD) Study among Japanese subjects (n = 719; mean age 72 years) showed that vitamin K intake assessed through Brief Dietary History Questionnaire was the only nutrient associated with a lower OA grade (KL grade < 2), which remained significant after sub-analysis based on sex [31]. In the subsequent ROAD study, a computer-aided diagnostic system was used to analyse minimum joint space (mJSW) width and osteophyte area of the knee joint. Vitamin K intake assessed through the Brief Dietary History Questionnaire was found to associated with mJSW but not the osteophyte area in the overall subjects. Sub-analysis based on sex showed that vitamin K, B1, B2, B6, and C were associated with mJSW, while vitamin E, B1, B2, niacin, and B6 were associated with osteophyte area. The relationship between vitamin K and knee OA parameters was not significant in men [32].

The progression of OA could also be monitored using circulating cartilage metabolism markers. Naito et al. [33] reported a significant positive correlation between serum undercarboxylated osteocalcin with serum bone metabolism markers (N-terminal telopeptide and bone alkaline phosphatase) and synovitis marker (hyaluronan), but not with cartilage metabolism markers (urinary C-telopeptide of type II collagen and C-terminal type II procollagen peptide) among Japanese patients with KL OA grade 3 or 4 (n = 25; mean age 76 ± 7.8 years). However, it should be noted that the sample collection of this study was not timed and the subjects did not fast, so the variability could be high. In the Health Aging and Body Composition Study (Health ABC) study involving 791 community-dwelling elderly (mean age 74 ± 3 years, 40% African American), a higher plasma uncarboxylated MGP level was associated with the occurrence of meniscus damage, osteophytes, bone marrow lesions, and subarticular cyst but not knee pain. Sub-analysis based on race showed that low uncarboxylated MGP was associated with articular damage in African Americans, while non-detectable plasma vitamin K was associated with meniscal damage, among Caucasians [34]. In terms of mobility, the Health ABC study found that circulating vitamin K level (< 0.5 nmol/L) was associated with mobility limitation and disability, compared to those with a level > 1.0 nmol/L. The uncarboxylated MGP was not associated with both mobility variables [35].

Two cross-sectional studies examined the relationship between OA and genetic MGP polymorphism. Secondary data analysis from a clinical trial (n = 376, mean age 71 ± 5.5 years) found no association between serum MGP level with hand OA [20]. Homozygote rs1800802 minor allele (GG) was shown to be associated with a lower risk of hand OA, compared to having 1 major allele at this locus. The same allele was associated with a lower risk of joint space narrowing and osteophyte formation. Homozygote rs4236 major allele (TT) was associated with a lower risk of joint space narrowing, compared to genotypes, with at least one minor allele. However, the sample size of this study was small and carboxylation of MGP was not considered [20]. In a genome-wide association study involving 12,784 subjects, a positive significant association between MGP variants (rs4764133) and hand OA was found but it was not linked with hip and knee OA [36].

The findings of cross-sectional studies are summarized in Table 2. 

### 2.3. Prospective Studies

Compared to the cross-sectional studies, prospective cohort studies would offer better insights into the relationship between vitamin K status and OA. In the Multicenter Osteoarthritis Study (MOST) involving 1180 subjects (mean age 62 ± 8 years), subclinical vitamin K deficiency (indicated by plasma phylloquinone level) at baseline was associated with incident radiographic knee OA and cartilage lesion but not osteophytes. It was also associated positively with OA at one or both knees. The association was adjusted with vitamin D but not lifestyle factors and other nutrients, which might also contribute to OA. In the Health ABC Study, plasma vitamin K level below detection at baseline predicted articular cartilage and meniscus damage among community-dwelling elderly after three years [34]. Similarly, El-Brashy et al. [25] showed that baseline vitamin K deficiency (as indicated by the plasma phylloquinone level) was associated with significantly decreased cartilage thickness at medial, lateral, and sulcus condyles, as revealed by ultrasound assessment, as well as OA progression. 

Subsequent analysis of the Health ABC study showed that subjects (n = 1323; mean age 74.2 ± 2.8 years) with plasma phylloquinone level < 0.5 nmol/L were more likely to develop mobility limitations and disability, compared to those with at least 1.0 nmol/L plasma phylloquinone. On the other hand, plasma uncarboxylated MGP (n = 716) was associated with mobility disability in a non-linear fashion but was not associated with mobility limitation [35]. Shea et al. also analysed the prospective data of Health ABC and Osteoarthritis Initiatives and found that adequate circulating phylloquinone level (≥ 1 nmol/L) and 25-hydroxyvitamin D (≥ 50 nmol/L) predicted better short physical performance battery scores and gait speed, in follow-up investigations. However, both nutrients were not associated with gait speed. On the other hand, adequate vitamin K (≥ 90 µg/day for women or 120 µg/day for men) and D intake (≥ 600 IU for age < 70 years, ≥ 800 IU for age ≥ 70 years) were associated with overall 20-m gait speed and chair stand completion time, on follow-up, but not in the 400-m walk time [37]. In another study, subjects with vitamin K deficiency also showed higher pain scores on The Western Ontario McMaster Scale Scores, after 12 months follow-up study [25].

In the Health ABC study involving 791 community-dwelling elderly (mean age 74 ± 3 years, 40% African American), a higher plasma uncarboxylated MGP level was associated with the occurrence of meniscus damage, osteophytes, bone marrow lesions, and subarticular cyst, but not knee pain. Sub-analysis based on race showed that low uncarboxylated MGP was associated with articular damage in African Americans, while non-detectable plasma vitamin K was associated with meniscal damage, among Caucasians [34]. In terms of mobility, the Health ABC study found that circulating vitamin K level (< 0.5 nmol/L) was associated with mobility limitation and disability, compared to those with a level > 1.0 nmol/L. The uncarboxylated MGP was not associated with both mobility variables [35].

The findings of prospective studies are summarized in Table 3. 

### 2.4. Clinical Trial

A clinical trial was attempted to study the effects of vitamin K on OA patients. In the study by Neogi et al. [39], 500 μg phylloquinone for 3 years did not alter the occurrence of hand OA, joint space narrowing, and osteophytes. Sub-group analysis reported that patients with vitamin K insufficiency (≤ 1nM) at baseline, who achieved sufficiency after treatment, showed a 47% reduction in the occurrence of joint space narrowing but not radiographic OA and osteophytes. The authors postulated that since both the treatment and control group received vitamin D supplementation, which could help to prevent OA, the difference was attenuated. The study also highlighted that those with vitamin K insufficiency, who were at greater risk to develop OA as illustrated in the previous sections, received that greatest benefits.

## 3. Mechanism of Action

Calcium crystals in the joint induced proliferation of synoviocytes and release of inflammatory cytokines and matrix metalloproteinases. Mineralization also led to cartilage stiffness and weakened the function of cartilage as a shock-absorbent [40]. Thus, joint calcification could initiate and exacerbate the progression of OA, and presented as a target for the action of vitamin K.

Carboxylation of matrix proteins, which is a functional activity of vitamin K, has been investigated as the action of vitamin K on chondrocytes. Chondrocytes from organ donors with OA produced mainly uncarboxylated MGP vesicles with fetuin per se, while chondrocytes from normal donors produced fully carboxylated MGP and fetuin complexes [41]. MGP can inhibit BMP-2 and -4 [42,43], thereby, suppressing calcification of the joint and formation of osteophyte. Alpha-fetuin acts as a carrier of MGP because it binds with fully carboxylated MGP and can be taken up by the chondrocytes through endocytosis [41].

Since OA is a localized inflammatory event of the joint, protein carboxylation activities of vitamin K could prevent inflammation and prevent the progression of OA. Indirect evidence from chondrocytes of rabbit with OA transfected with γ-glutamyl carboxylase generated increased type II collagen, which is indicative of good chondrocyte survival and differentiation, and lower metalloproteinase and type X collagen, which is indicative of chondrocyte apoptosis. Protein and gene expressions of tumor necrosis factor and interleukin-1 beta were also decreased [44]. GRP is a vitamin K-dependent protein expressed in the soft tissues of humans and involved in pathological calcification [45]. As mentioned previously, cartilage calcification is a pathological feature of OA [40]. Initial assessment by Cavaco et al. [46] showed that treatment with undercarboxylated and carboxylated GRP inhibited inflammation of chondrocytes and synoviocytes stimulated with interleukin-1β, as evidenced by lower cyclooxygenase II, metalloproteinase 13, and prostaglandin E-2 expression [46]. Treatment with undercarboxylated and carboxylated GRP was shown to reduce tumor necrosis factor-alpha and prostaglandin E-2 levels secreted by lipopolysaccharide- and hydroxyapatite-stimulated THP-1 monocytes. THP-1 cells overexpressing GRP also showed lower gene expression of interleukin-1β, nuclear factor kappa-light-chain-enhancer of activated B cells, and tumor necrosis factor-alpha when challenged with lipopolysaccharide or hydroxyapatite [47]. Thus, this study showed that protein carboxylation might not affect the anti-inflammatory activities of GRP. Yet, THP-1 might not reflect the normal functions of primary macrophages and monocytes because they are immortalized. 

The mechanism of actions of vitamin K in preventing OA is shown in Figure 1. 

## 4. Perspectives

Currently, the recommended dietary allowance or RDA of vitamin K is not established due to insufficient data. Therefore, average intake values (AI) derived from healthy representative populations have been used. The AI of vitamin K for adult men and women are 120 and 90 μg/day [23]. Although deemed to be rare, a recent report from the Prevention of Renal and Vascular End-Stage Disease (PREVEND) Study reported that vitamin K deficiency, defined through plasma desphospho-uncarboxylated MGP level (>500 pmol/L), was found in 31% of the total subjects (n = 4275 subjects, aged 53 ± 12 years) [48]. Vitamin K deficiency bleeding is a serious condition found among infants but largely curbed in developed countries, through prophylaxis [49]. Vitamin K supplementation is not associated with serious health effects, thus, its tolerable upper intake level has not been fixed [50]. Therefore, long-term supplementation of vitamin K for preventing OA might be safe, but there is a paucity of trial data to support this claim. The effective dose of vitamin K in preventing OA remains elusive, but some data suggest that maintaining the circulating vitamin K concentration around 1.0 nmol/L is associated with better joint health [29,37].

However, vitamin K showed interaction with drugs, notably warfarin (an anticoagulant) and nutrients, for instance, vitamin E. Warfarin functions by suppressing hepatic vitamin K-epoxide reductase, which in turns inhibits the productions of vitamin K-dependent plasma clotting factors [51]. Patients receiving warfarin are commonly advised to avoid taking vitamin K-rich food, like green leafy vegetables. A recent meta-analysis showed that interference might only occur at very high vitamin K intake (>150 μg/day) [52]. The single clinical trial of vitamin K on OA supplemented the subjects with 500 μg/day, so this interaction should be cautioned [39]. However, there is a lack of evidence showing that normal dietary vitamin K level would interfere with the function of warfarin [52].

Vitamin E supplementation has been reported to increase the risk of hemorrhagic stroke [53]. This mechanism might be linked to its antagonistic action on vitamin K metabolism. Rats supplemented with α-tocopherol (0–500 mg/kg diet) dose-dependently reduced the level of phylloquinone but not menaquinone, in multiple organs except serum and liver [54]. In a human trial, 12-week supplementation of a-tocopherol (1000 IU daily) resulted in a higher under-γ-carboxylated prothrombin level, indicating a poorer vitamin K status in subjects with a normal coagulation profile [55]. Therefore, vitamin E, notably a-tocopherol, could interfere with the joint protective effects of vitamin K. However, we previously reviewed that vitamin E itself could protect against osteoarthritis [10]. Thus, the sum of the effects of these interactions on joint health awaits further studies. Although this review focuses on the effects of vitamin K on joint, it should be noted that the human diet contains numerous nutrients, and they could contribute synergistically towards joint protection. This is illustrated by the ROAD study, whereby intake of vitamin B, C, E, and K were reported to be associated with different aspects of joint health [32]. 

Although most of the discussion in this review focused on the cartilage, it should be noted that joint consists of multiple tissues, such as tendon, synovium, and subchondral bone. The effects of vitamin K on bone and osteoporosis have been extensively studied, whereby it promotes the carboxylation of osteocalcin, leading to mineral accretion and differentiation of osteoblasts, while decreasing the formation of osteoclasts [56]. However, it is uncertain that this would contribute to a better loading capacity of the subchondral bone, leading to OA prevention. We also discussed that GRP could suppress the inflammation cascade in THP-1 cells, but it is independent of the carboxylation status (or indirectly the vitamin K status) of GRP [47]. Other than that, the effects of vitamin K on other tissues of the joints are less studied. This research gap should be bridged in future experiments.

## 5. Conclusions

Human observational studies show that vitamin K could prevent OA but evidence from clinical trials is limited. Some important questions concerning the intake level, dose of supplementation, and vitamin K type that is the most effective against OA awaits further studies. At this juncture, maintaining an adequate intake of vitamin K seems to be the best recommendation for elderly who wish to prevent OA.

## Figures and Tables

**Figure 1 nutrients-12-01208-f001:**
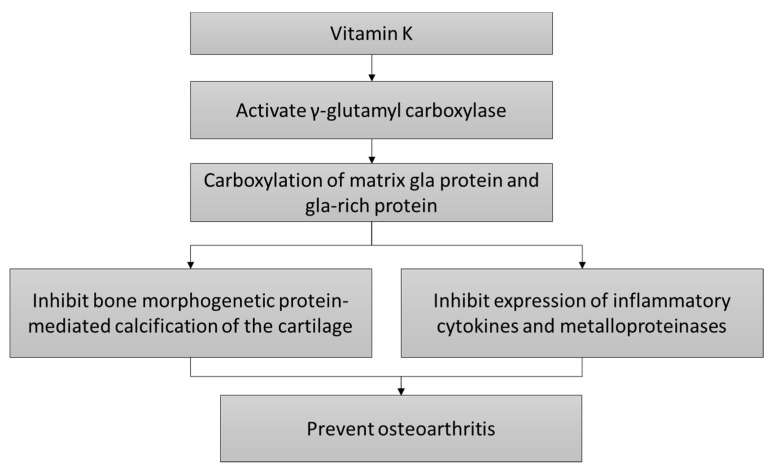
The cartilage protective mechanism of vitamin K.

**Table 1 nutrients-12-01208-t001:** The relationship between vitamin K status and OA from case-control studies.

Reference	Study Design	Outcomes
Roberts et al. 1996 [24]	Patients with femoral neck fracture (*n* = 13): 10 women, median age 80 years; 3 men, median age 85 years. OA patients (*n* = 16): 10 women, median age 71 years; 6 men, median age 64 years. Healthy volunteers (*n* = 25): 16 women and 9 men, median age 77 years. Blood was sampled from the patients before and after surgery. Serum vitamin K1 measured by HPLC.	Vitamin K1: ↓ in patients with femoral neck fracture and OA vs. normal controls; ↓ after surgery. Carboxylated and undercarboxylated osteocalcin: ~ between case and control.
Silaghi et al. 2012 [28]	Non-inflammatory arthritis group: *n* = 17, median age 55 years, patients with OA or chondrocalcinosis Inflammatory arthritis group: *n* = 9, median age 55 years, patients with juvenile idiopathic arthritis, ankylosing spondylitis Control: *n* = 30, median age 42 years, no radiographic evidence of arthritis Uncarboxylated MGP measured by ELISA	Serum uncarboxylated MGP: ↑ in the controls vs. patients with arthritis; ↑ in the non-inflammatory arthritis group vs. inflammatory arthritis group. Synovial uncarboxylated MGP: ↑ in the inflammatory arthritis group vs. non-inflammatory arthritis group Synovial/serum uncarboxylated MGP ratio: ↑ in the inflammatory group vs. non-inflammatory group.
Rafael et al. 2014 [26]	OA samples from patients undergoing total knee replacement surgery. Control samples from the autopsy of subjects with no history of joint disease. Immunostaining technique.	Undercarboxylated GRP: ↑ in the OA cartilage matrix and synovial membrane vs. to carboxylated one. Carboxylated MGP: ↑ in the control cartilage matrix and synovial membrane vs. undercarboxylated ones.
Bing and Feng 2015 [27]	Case: 178 knee OA patients, aged 62.8 ± 7.4 years, 114 women: 64 men Control: 160 healthy outpatients in a hospital, aged 63.2 ± 8 years, 94 women: 66 men OA assessment: knee radiographs with KL criteria (at least ≥ 2 at one knee to be selected as cases) Uncarboxylated MGP measured using ELISA	Serum uncarboxylated MGP: ↓ in the OA patients vs. control. Synovial fluid uncarboxylated MGP level correlated negatively with the disease severity among the OA patients.
El-Brashy et al. 2016 [25]	Case: 40 knee OA patients (36 women: 10 men), age 50.4 ± 4.9 years, KL grade 2 or less Control 20 healthy individuals (85 women: 15 men), age 48.9 ± 4.6 years, Plasma phylloquinone level was measured using ELISA OA assessed using pain scale, The Western Ontario McMaster Scale (WOMAC), Thomas Grading Score and musculoskeletal ultrasound	Plasma vitamin K: ↓ in knee OA patients vs. control WOMAC score: ↑ in patients with vitamin K deficiency vs those who were sufficient. A significant positive correlation was found between plasma vitamin K level and medial cartilage thickness.

Abbreviation: ELISA, enzyme-linked immunosorbent assay; HPLC, high-performance liquid chromatography; GRP, gla-rich protein; MGP, matrix gla protein; OA, osteoarthritis; vs., versus; ↓, decrease; ↑, increase; ~, similar.

**Table 2 nutrients-12-01208-t002:** The relationship between vitamin K status and osteoarthritis (OA) from cross-sectional studies.

Reference	Study Design	Findings
Neogi et al. 2006 [29]	Framingham Offspring Study. Subjects: *n* = 672, 53% women. Mean age 66 years OA assessed using Framingham OA atlas. Phylloquinone level assessed using HPLC.	↓ hand OA and large osteophyte for subjects with the Q4 (1.81–21.5 nmol/L) of phylloquinone level vs. Q2 (0.59–1.02 nmol/L) & Q1 (0.05–0.58 nmol/L). ↓ joint space narrowing for hand OA in Q4 vs. Q1. A threshold effect of 1 nmol/L was observed, in which ↓ OA prevalence after that level. Knee joint OA not associated with phylloquinone level
Oka et al. 2009 [31]	Research on Osteoarthritis Against Disease (ROAD) study. Subjects: 719 Japanese subjects, 62.4% women. Mean age 72.1 ± 6.3 years for men, 72.0 ± 7 yrs for women. OA assessed through radiographs using KL grade. Vitamin K intake assessed through Brief Dietary History Questionnaire	Vitamin K intake was associated with ↓ OA grade ≥ 2 (OR 0.75 95% CI 0.63–0.89) or ≥ 3 (OR 0.67 95% CI 0.53–0.84). Based on sex, the relationship of OA grade ≥ 2 remained for men (OR 0.76 95% CI 0.59–0.95) and women (OR 0.74 95% CI 0.58–0.96). For women, the relationship of OA grade ≥ 3 (OR 0.61 95% CI 0.45–0.81) was associated with vitamin K intake.
Misra et al. 2011 [20]	376 subjects participated in a randomized controlled trial on the effects of vitamin K supplementation on bone and vascular health. 59% women, age 71 ± 5.5 years.	No association between serum MGP levels and hand OA. Homozygote rs1800802 minor allele (GG) was associated with a lower risk of hand OA vs. having 1 major allele at this locus (OR 0.56, 95% CI 0.32–0.99). The same allele was associated with a lower risk of joint space narrowing (OR 0.25, 95% CI 0.11–0.58) and osteophyte formation (OR 0.31, 95% CI 0.17–0.58). Homozygote rs4236 major allele (TT) was associated with a lower risk of joint space narrowing compared to genotypes with at least one minor allele (OR 0.52, 95% CI 0.35–0.78).
Naito et al. 2012 [33]	25 patients with KL grade 3/4 for bilateral knee OA (age: 76 + 7.8 years, BMI; 24.9 + 4.7) were recruited. Serum undercarboxylated osteocalcin was measured by ELISA.	Serum undercarboxylated osteocalcin: marginal ↑ increase in patients with KL grade 6/7/8. Positive correlation between undercarboxylated osteocalcin and bone metabolism markers (serum N-terminal telopeptide and bone alkaline phosphatase) and synovitis marker (serum hyaluronan), but not with cartilage metabolism markers (urinary CTX-II and C-terminal type II procollagen peptide).
Ishii et al. 2013 [30]	58 bones from patients (13 men; 45 women) undergoing (aged 73 ± 8 years) total knee arthroplasty (all grade 4 OA) were collected. Vitamin K2 in the medial and lateral femoral and tibial condyles were compared. Vitamin K2 was analysed using HPLC.	↑ Vitamin K2 in the lateral femoral and tibial condyles than the medial condyles; ↑ in the femoral lateral/medial condyles than the tibial lateral/medial condyles but only the lateral parts were significant. The difference in age and sex was not significant.
Muraki et al. 2014 [32]	Research on Osteoarthritis/Osteoporosis Against Disability Study (ROAD) Subjects: 827 subjects (305 men, 522 women), mean age 69.2 ± 9.3 years. A knee OA computer-aided diagnostic system was used to analyse minimum joint space width (mJSW) and osteophyte area. Vitamin K intake (previous month) assessed via Brief Dietary History Questionnaire	Vitamin K intake was significantly associated with mJSW but not osteophyte area. In women, vitamin K, B1, B2, B6, and C intakes were associated with mJSW, after adjusting for age, BMI and total energy. In women, vitamin E, B1, B2, niacin, and B6 were associated with osteophyte area.
Shea et al. 2015 [34]	791 community-dwelling elderly (mean age 74+3 years, BMI 27.7 + 4.8, 67% women) from Healthy, Aging and Body Composition Study (Health ABC). 40% African Americans and 60% Caucasians. Median follow up period of 37 months. OA assessed via Magnetic resonance imaging on both knees Plasma phylloquinone measured using reversed-phase HPLC Vitamin K intake assessed via The Health ABC food frequency questionnaire MGP assessed via ELISA	↑ plasma dephosphorylated uncarboxylated MGP (highest vs lowest quartiles) was associated with ↑ risk of meniscus damage (OR: 1.6, 95%CI 1.1–2.3), osteophytes (OR: 1.7, 95%CI 1.1–2.5), bone marrow lesions (OR: 1.9, 95%CI 1.3–2.8) and subarticular cyst (OR: 1.5, 95%CI 1.0–2.1). African Americans in the lowest uncarboxylated MGP quartile had the highest risk of having articular damage. Caucasians with non-detectable plasma vitamin K were more likely to have meniscal damage.
Shea et al. 2019 [35]	Health ABC: *n* = 1323 (635 men) for plasma phylloquinone HPLC measurement; *n* = 716 for uncarboxylated MGP measurement. Age 74.2 ± 2.8 years, 40% black/60% white. Mobility limitation defined as 2 consecutive semiannual reports of difficulties either walking 1/4 miles or climbing 10 steps. Mobility disability defined as 2 consecutive semiannual reports of huge difficulties/inability to walk 1/4 miles.	Subjects with circulating vitamin K level < 0.5 nmol/L had more risk for mobility limitation (OR: 1.49 (96% CI 1.04–2.13)) and disability (OR: 1.95 (96% CI 1.08–3.54)) compared to those with > 1.0 nmol/L uncarboxylated MGP was not associated with both variables.

Abbreviation: CI, confidence interval; ELISA, enzyme-linked immunosorbent assay; HPLC, high-performance liquid chromatography; GRP, gla-rich protein; KL, Kellgren and Lawrence; MGP, matrix gla protein; OA, osteoarthritis; OR: odds ratio; Q, quartile; ↓, decrease; ↑, increase.

**Table 3 nutrients-12-01208-t003:** The relationship between the vitamin K status and OA from prospective studies.

Reference	Study Design	Findings
Misra et al. 2013 [38]	1180 subjects from Multicenter Osteoarthritis (MOST) study, 62% women, mean age 62 + 8 years. Knee radiographs or MRI scan obtained at baseline and 30 months later. Plasma phylloquinone measured at baseline using HPLC. New incidence of OA referred to KL grade 0/1 to > 2, cartilage lesion based on Whole-Organ Magnetic Resonance Imaging Score from 0 > 1 and osteophytes from 0/1 to >2.	Subclinical vitamin K deficiency (9.2% of the overall population) was associated with incident radiographic knee OA (RR: 1.56 95% CI 1.08–2.25) cartilage lesion (RR: 2.39 95% CI 1.05–5.40) but not with osteophytes (RR 2.35 95%CI 0.54–10.13). It was also associated with OA at one or both knees (RR 1.33 95% CI 1.01–1.75 and RR 2.12 95% CI 1.06–4.24).
Shea et al. 2015 [34]	Cross-sectional and longitudinal studies. 791 community-dwelling elderly (mean age 74 + 3 years, BMI 27.7 + 4.8, 67% W) From Healthy, Aging and Body Composition Study (Health ABC). 40% African Americans and 60% Caucasians. Median follow up period of 37 months. OA assessed using magnetic resonance imaging on both knees. Vitamin K assessed using plasma phylloquinone measured using reversed-phase HPLC. Dietary intake assessed using the Health ABC food frequency questionnaire. MGP assessed using ELISA.	Non-detectable plasma vitamin K level (<0.2 nmol/L) at baseline predicted articular cartilage (OR: 1.7, 95%CI 1.0–3.0) and meniscus damage (OR 2.6, 95%CI 1.3–5.2) after three years, compared to those with sufficient vitamin K. Plasma uncarboxylated MGP did not predict knee pain at baseline.
El-Brashy et al. 2016 [25]	A case-control study with 12 months longitudinal follow up. Case: 40 knee OA patients (36 women: 10 men), age 50.4 ± 4.9 years, KL grade 2 or less Control 20 healthy individuals (85 women: 15 men), age 48.9 ± 4.6 years Plasma phylloquinone level was measured using ELISA. OA assessed using pain scale, The Western Ontario McMaster Scale (WOMAC), Thomas Grading Score and musculoskeletal ultrasound.	↑ WOMAC score and pain scale of patients with deficiency vs. those with sufficient level. ↓ cartilage thickness at medial, lateral and sulcus condyles in patients with vitamin K deficiency. The same observation was obtained for Thomas score at the medial compartment and total score. Vitamin K deficiency (0.5 nmol/L) among the patients was associated with radiographic OA progression (RR: 2.08, 95% CI 1.30–3.32). The best cut-off for vitamin K on radiographic OA progression was 1.74 nmol/L, on ultrasound was 1.28 nmol/L.
Shea et al. 2018 [37]	Two prospective cohort studies: Health, ABC and Osteoarthritis Initiative (OAI). In Health ABC: *n* = 1069, 60% women, aged 75 ± 3 years. Plasma phylloquinone and 25-hydroxyvitamin D were determined. Physical function: short physical performance battery and usual 20-meter gait speed. In OAI: n = 4475, 58% women, aged 61 ± 9 yrs. Vitamin K and D intakes were determined using the Block Brief 2000 food frequency questionnaire. Physical function: 20-meter gait speed and chair stand completion time. Follow-up period for both: 4–5 years.	In Health ABC, adequate circulating K (≥ 1 nmol/L) and D (≥ 50 nmol/L) predicted better physical performance battery scores and gait speed on follow up. Changes in physical performance score were not associated with vitamin D status. Both variables showed no correlation with gait speed. In OAI, adequate vitamin K (≥ 90 µg/day for women or 120 µg/day for men) and D intake (≥ 600 IU for age < 70 years, ≥ 800 IU for age ≥ 70 years) were associated with overall 20-m gait speed and chair stand completion time on follow up, but not 400-meter walk time.
Shea et al. 2019 [35]	Health ABC. *n* = 1323 (635 men) for plasma phylloquinone HPLC measurement, *n* = 716 for uncarboxylated MGP measurement. Age 74.2 ± 2.8 years, 40% black/60% white. Mobility limitation defined as 2 consecutive semiannual reports of difficulties either walking 1/4 miles or climbing 10 steps. Mobility disability defined as 2 consecutive semiannual reports of huge difficulties/inability to walk 1/4 miles. Median follow-up: 6.4 (8.6) years for limitation and 10.3 (5.8) years for disability.	Subjects with circulating vitamin K level <0.5 nmol/L were more likely to develop mobility limitation (OR: 1.27 (96% CI 1.05–1.53)) and disability (OR: 1.34 (96% CI 1.01–1.76)) compared to those with 1.0 nmol/L. After adjustment for knee pain, the association with disability was attenuated significantly (OR: 1.26 (96% CI 0.96–1.67)). Plasma uncarboxylated MGP was not associated with mobility limitation but was associated with incident mobility disability non-linearly (n-shape).

Abbreviation: CI, confidence interval; ELISA, enzyme-linked immunosorbent assay; HPLC, high-performance liquid chromatography; KL, Kellgren and Lawrence; MGP, matrix gla protein; OA, osteoarthritis; OR, odd rations; RR, relative risk; yrs, years; ↓, decrease; ↑, increase.

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
