# Peer review of "The Relationship between Vitamin K and Osteoarthritis: A Review of Current Evidence"

_nutrients, 2020, doi:10.3390/nu12051208_

Round 1

Reviewer 1 Report

This is a well-documented, detailed review on the current evidence of the role of Vitamin K and OA. 

It is well written and concisely critiques current trials with vitamin K.

There are only a couple of comments I have on the review.

Firstly, OA is a slow progressing disease which also is attributed to mechanical wear and tear injury to the cartilage. I may have missed it in the review, but it did not come across how the vitamin K could offset the mechanical processes leading to OA. In Figure 1 the vitamin K shows activation of Y- glutamyl carboxylase and subsequent carboxylation of the matrix gla-rich protein which inhibits BMP-2 and inflammatory cytokines. How does this fit in with mechanical damage associated with OA.

On Page 12 line 273 "However, we previously reviewed that vitamin E itself could protect against osteoarthritis 273 [10]. This, the sum of the effects of these interactions on joint health awaits further studies. Although this review focuses on the effects of vitamin E on joint, it should be noted that the human diet contains numerous nutrients, and they could contribute synergistically towards joint protection."  When the author states "...this review focuses on the effects of vitamin E do they mean this review or are the referring to reference 10? Just need to clarify.

Author Response

Dear reviewer, 

Thank you for reviewing our manuscript. We appreciate the constructive comments provided and they are addressed in the attached response sheet. 

Thank you. 

Reviewer 2 Report

In this paper, the author provides a comprehensive review on the relationship between vitamin K and osteoarthritis. The literature is discussed in a critical manner and this paper provides a useful resource for those in the field. This review is very well written and the data are clearly presented with no overstatements.

Minor comments:

Tables. The heading of column 1 could read “Reference” instead of “Researchers”

Table 2. For the Shea MK 2015 entry, the author could consider rewording the “food frequency questionnaire (24 hours recall)” entry; as reported by the cited paper, the “FFQ food list was derived from the 24 hrs recall data” (Shea MK, 2015). This may avoid potential misunderstanding for those readers who may not be familiar with collection of dietary information.

Author Response

(The authors gave the same response as above.)

Reviewer 3 Report

The introduction describes all the parameters that have influence in OA as vit D,vit E,polyphenols.The role of vit K is actually undefined and may be only supposed by some experiments and some analyses well reported as the case control studies,all reported in table 1,cross sectional studies,all reported in table 2 and prosecutive reported in table 3.Well described the mechanism of action of vit.K in OA mainly through GRP that is expressed in condrocytes and knockout animals develop calcification of blood vessels.The review is very exhaustive,highly useful for the readers of the field,very useful the summaries presented of each paper, all the references of the topic are reported.I found only one blunder at line 185:ug instead of micro in greek.

Author Response

Dear reviewer, 

Thank you for reviewing our manuscript. We appreciate the constructive comments provided and they are addressed in the attached response sheet. 

Thank you. 

This manuscript is a resubmission of an earlier submission. The following is a list of the peer review reports and author responses from that submission.